# Diguanylate Cyclase (DGC) Implicated in the Synthesis of Multiple Bacteriocins via the Flagellar-Type III Secretion System Produced by *Pectobacterium carotovorum* subsp. *carotovorum*

**DOI:** 10.3390/ijms23105649

**Published:** 2022-05-18

**Authors:** Ruchi Briam James Sersenia Lagitnay, Han-Ling Chen, Yen-Chun Chen, Duen-Yau Chuang

**Affiliations:** 1College of Arts & Sciences, Bayombong Campus, Nueva Vizcaya State University, Bayombong 3700, Nueva Vizcaya, Philippines; rslagitnay@nvsu.edu.ph; 2Department of Chemistry, National Chung Hsing University, Taichung City 400, Taiwan; hanling@smail.nchu.edu.tw (H.-L.C.); yenchun2007@gmail.com (Y.-C.C.)

**Keywords:** diguanylate cyclase, bacteriocin, flagellar-type III secretion system, *Pectobacterium carotovorum* subsp. *carotovorum*

## Abstract

The plant pathogen *Pectobacterium carotovorum* subsp. *carotovorum* (previously *Erwinia carotovora* subsp. *carotovora*) causes soft rot and stem rot diseases in a variety of crops, including Chinese cabbage, potato, and tomato. The flagellar-type III secretion systems were used by *Pcc*’s virulence mechanism to export proteins or bacteriocins to the outside of the cell. DGC, a virulence factor that cyclizes c-di-GMP, a common secondary signal in physiological processes and toxin control systems of many bacteria, was discovered in *Pcc*’s genomic DNA. The *dgc* gene in *Pcc* was blocked using the method of homologous recombination in our study. In the in vivo setting, the results demonstrated that the *dgc* knockout strain does not release low molecular weight bacteriocins. The bacteriocin gene (*carocin S2*, *carocin S3*, *carocin S4*) and the flagellar-type III secretion system genes were also unable to be transcribed by the *dgc* knockout strain in the transcription experiment. We also observed that the amount of bacteriocin expressed changed when the amount of L-glutamine in the environment exceeded a particular level. These data suggested that L-glutamine influenced physiological processes in *Pcc* strains in some way. We hypothesized a relationship between *dgc* and the genes involved in *Pcc* LMWB external export via the flagellar-type secretion system based on these findings. In this study, the current findings led us to propose a mechanism in which DGC’s cyclic di-GMP might bind to receptor proteins and positively regulate bacteriocin transcription as well as the synthesis, mobility, and transport of toxins.

## 1. Introduction

When bacteria are exposed to various environmental conditions, they develop a number of transmission pathways in their cells that produce physiological changes involving a wide range of proteins and chemicals. The cyclic diguanylate monophosphate (c-di-GMP), which connects external stimuli to cells in signal transmission, is a common and important secondary signaling component in many bacterial species, including *Pectobacterium carotovorum* subsp. *carotovorum* (*Pcc*) [1,2,3,4]. A class of enzymes known as diguanylate cyclase (DGC) with a conserved GGDEF motif synthesizes this twofold symmetric molecule at the intracellular level from two molecules of guanosine triphosphate (GTP) [5,6]. Diguanylate cyclase has been demonstrated to control a range of cell activities by producing c-di-GMP. Cell cycle regulation, adhesion, biofilms, motility, toxicity, and cellulase modulation are all regulated by changes in c-di-GMP concentrations [7,8,9]. To regulate a wide range of cellular processes, c-di-GMP must first bind to protein receptors and then interact with downstream receptors.

The PilZ protein domain, riboswitch [10,11], cAMP receptor CRP [12], and CLP [13] of the protein family are examples of c-di-GMP receptors. These receptors can interact directly with c-di-GMP. Recent findings suggest that c-di-GMP affects type VI secretion systems (T6SSs) [14] as well as T3SSs [15]. The action of c-di-GMP in T3SSs appears to be direct and mediated once again by a central ATPase, as the flagellar export ATPase FliI from a variety of distantly related bacteria has been found to bind to c-di-GMP [8,15] exclusively.

*Pectobacterium carotovorum* subsp. *carotovorum* (*Pcc*) secretes numerous pathogenicity-related proteins that cause soft-rot disease in a variety of plant species [16]. In the virulence mechanism of *Pcc*, multiple types of secretion systems (TSS) are used to export proteins or bacteriocins to the outside of the cell. Protein molecules that traverse the outer and inner membranes are found in the secretion system, and c-di-GMP can bind to certain receptor proteins to control secretion system gene transcription [17,18]. Our earlier findings point to a Type III secretion pathway for bacteriocin extracellular export that is a mix of injectisome and flagellar secretion [19].

Prior investigations have demonstrated that DGC can be regulated at either the level of enzymatic activity or the level of expression within the surrounding environment [20], despite the fact that intracellular transmission channels and regulatory factors are involved at several levels. This study will aid in a better understanding of the role of DGC in the regulation, transcription, and secretion of numerous bacteriocins by the *Pcc* via the mix T3SS.

## 2. Results

### 2.1. Isolation and Detection of Tn5 Insertion Mutants

When *Pcc* is exposed to rifampicin, it may undergo natural mutation and create progeny strains with rifampicin resistance genes. Even after mutations, these mutants continue to produce bacteriocins. We used rif-TO6, a rifampicin-resistant *Pcc* strain, as the recipient of bacterial conjugation and *E. coli* (1830) with the kanamycin resistance gene transposon Tn5 as the provider of conjugative reproduction in this study. Following mating, about 10,000 strains were tested on Drigalski’s and LB medium with antibiotics kanamycin and rifampicin. Following that, bacteriocin assays [21,22] were used to identify strains with successfully disrupted low-molecular-weight bacteriocin-related genes.

The size of the inhibitory zone around each isolate was compared to that of rif-TO6 in a bacteriocin assay. According to this evidence of mutation (Figure 1A), the absence of inhibitory zones confirmed that the mutant colonies’ transposon Tn5 had been inserted into LMW bacteriocin-related genes. The wild-type rif-TO6 and Tn5 insertional mutants that underwent bacteriocin testing verified that the strains are *Pcc*, according to the results from the Modified Drigalski medium (Figure 1B).

### 2.2. DNA Amplification at the Tn5 Insertion Junction

To investigate if Tn5 was incorporated into the genomic DNA of suspected isolates, we used a nested PCR using two primers, P3 and P4, to amplify the *nptII* resistance gene on Tn5 (about 500 bp in length). This PCR approach can be used to confirm whether the wild strain’s DNA has been inserted with transposon Tn5. All isolated putative mutants had a 500-bp band; however, because wild-type rif-TO6 lacked Tn5 insertion, there was no band (data not shown).

To identify Tn5-interrupted genes, genomic DNA from the various mutant strains was amplified with Thermal Asymmetric Interlaced PCR (TAIL-PCR) using an array of specific primers. The template DNA was extracted from *Pcc*’s insertional mutant’s genomic DNA, and the isolated chromosomal DNA was treated with *EcoRI* before being validated on a 0.7 percent agarose electrophoresis gel. Appendix A shows the results of the TAIL-PCR.

TAIL-PCR products with a length of 1070 bp were obtained and sequenced. DNASIS MAX 3.0 software was used to deduce the amino acid sequences from the putative mutants. Among the different insertion mutants, the TT6-6 DNA sequence revealed that this carries a diguanylate cyclase with a 30% homogeneity in *Shigella boydii*, according to results from the BLAST programs of the National Center for Biotechnology Information website (National Library of Medicine, Bethesda, MD, USA) Table 1.

### 2.3. Role of DGC on Pcc’s Bacteriocin Secretion

We used rifampicin-resistant *Pcc* 89-H-12 bacteria as the gene knockout strain to better understand the link between DGC and bacteriocin production.

Because the 89-H-12 strain can synthesize carocin S2, S3, and S4, the low-molecular-weight bacteriocins (Figure 2A), using it as a gene selection strain will help the researcher better understand *Pcc*’s virulence mechanism.

The pG-dgc-Kan and pG-dgc-Tet constructs are utilized for double confirmation and undergo homologous recombination. Because the *Pcc* 89-H-12 strain is unable to produce kanamycin- and tetracycline-resistant bacteria, these antibiotic gene fragments were used as a selection marker for mutant screening.

Two dgc defective mutants, 89-H-rif-12/dgc-Kan, and 89-H-rif-12/dgc-Tet were compared to the wild-types *Pcc* 3F3 and *Pcc* 89H-rif-12 on their ability to secrete various bacteriocins, whereas sp33 was used as an indicator strain.

After transformations, in vivo tests revealed that the knockout strains had no LMWB secretions, whereas the wild types had normal LMWB secretion at the cellular level (Figure 2B). Carocin S1, S2, S3, and S4 cannot be transported extracellularly when the dgc gene is knocked out.

### 2.4. Regulation of DGC on Pcc’s Bacteriocin Secretion

A transcriptional study was also conducted to see how bacteria generated, regulated, and secreted bacteriocin. The wild-type and dgc knockout (89-H-rif-12/dgc-Kan and 89H-rif-12/dgc-Tet) strains were used in this experiment, which grew exponentially after induction. These induced strains were then reincubated to allow the cells to manufacture bacteriocin before being collected to extract the bacteriocin’s RNA for in vitro testing. The ability of the induced cells to create, regulate, and release bacteriocin was studied using a two-step endpoint reverse transcriptase polymerase chain reaction (endpoint RT-PCR) [23,24]. The 16 RNA expression, which is highly conserved among bacteria species, was used as a control for comparing gene expression in this technique.

The three bacteriocin genes, *caroS2k*, *caroS3k*, and *caroS4k*, all lose their ability to express on an agarose gel electropherogram (Figure 3), indicating that the dgc gene regulates the bacteriocin toxin gene to prevent the strain from secretion. We have also identified certain regulatory genes (*crp*, *brg*, and *gln*h) involved in the manufacture of bacteriocin based on earlier research. These genes can still be transcribed, indicating that the upstream end of the *dgc* gene is unaffected.

According to the findings of this study, further deletion of *dgc* genes inhibits the production of *fliC* and *fliG*. Because the T3SS genes *fliC* and *fliG* are involved in the synthesis of chaperones necessary for the delivery of LMWBs, knocking down dgc in *Pcc* limits bacteriocin secretion to the extracellular environment.

### 2.5. Extracellular L-Glutamine Concentrations Have Varied Effects on 89-H-rif-12

Bacteria’s tremendous adaptability to the environment is based on their ability to detect changes in the environment. To explore bacteriocin secretion in a nutrient-rich environment, we cultured 89-H-rif-12 in an environment containing L-glutamine to see if it affected bacteriocin secretion and regulatory gene expression. So, we cultured 89-H-rif-12 in 2.5 mM, 5.0 mM, and 10.0 mM L-glutamine LB medium and the control group without L-glutamine and incubated them overnight at 28 °C.

The RNA was extracted from exponentially grown cells induced at various L-glutamine concentrations for a two-step endpoint RT PCR analysis of genes associated with the carocin genes, regulatory genes, and type III secretory genes in both the wild-type and the dgc defective mutant.

The ability of the induced cells to create, regulate, and release bacteriocin was studied using a two-step endpoint reverse transcriptase polymerase chain reaction (endpoint RT-PCR). The gene segments 16s RNA, *dgc*, *crp*, *brg*, *glnH*, *flhA*, *fliC, fliG*, *caroS2k*, *caroS3k*, and *caroS4k* were successively amplified (Figure 4A).

Based on Figure 4B, there is no significant difference in the expression levels of *brg* and *glnH* in the L-glutamine environment. The expression levels of related protein genes *dgc*, *crp*, and the third secretion system-related protein genes *flhA*, *fliC*, and *fliG* were significantly increased compared to the control group, despite the fact that GlnH is an active transport protein that transports extracellular L-glutamine into the cell.

Increased production of these proteins associated with bacteriocin regulation also resulted in increased expression of the *caroS2k*, *caroS3k*, and *caroS4k* bacteriocin genes, proving that there is a causal relationship between bacteriocin expression and L-glutamine concentration.

## 3. Discussion

A number of transmission pathways are created by rapidly altering genes in cells in order to boost the ability of bacteria to compete for survival in the natural environment. The transmission pathway involves a large number of proteins and compounds. The secondary signal produced by DGC protein has been implicated in prior research, and c-di-GMP is required for the regulation of *Pcc*’s bacteriocin synthesis mechanism.

In addition to bacteriocin secretion, the regulatory mechanism for delivering bacteriocin to the extracellular environment comprises secreted proteins and immune proteins. In this study, blocking the *dgc* gene of 89-H-rif-12 resulted in no zone of inhibition around the mutant strains in the bacteriocin assay, indicating the strain’s inability to produce low-molecular-weight bacteriocin, and the related gene was turned off to prevent the production of bacteriocin. These findings clearly show how c-di-GMP, a secondary signal produced by DGC proteins, affects the production of bacteriocins.

DGC positively regulates the expression of *fliC* and *fliG* of the type III secretion system but cannot regulate *flhA*, despite the fact that blocking the T3SS prevents the secretion of bacteriocins into cells. Furthermore, the *dgc* defective mutant cannot transcribe bacteriocin toxin genes such as *caros2k*, *caros3k*, and *caros4k*. As a result, it is thought that DGC not only transports toxins in the *Pcc* regulatory system but also plays a role in the manufacture of bacteriocin.

Bacteria’s remarkable environmental adaptability is dependent on their ability to detect changes in their environment. The probable mechanism (Figure 5) of DGC regulating the bacteriocin gene is hypothesized based on the relevant literature.

DGC responds to UV irradiation and nutrient-rich environments, such as high glutamine concentrations. The signal created by cAMP and CRP changes the conformation of DGC when the sensor protein on bacteria is triggered by the outside world. DGC cyclizes c-di-GMP, and c-di-GMP generates a compound with CLP to remove CLP from the bacteriocin promoter-binding site; then, DGC activates the DNA-binding site and binds to the DGC promoter sequence to induce *dgc* transcription. It is released so that the bacteriocin promoter can be coupled with RNA polymerase to create bacteriocin toxin and immune proteins during transcription.

*DGC* was also observed to have an effect on the regulation of genes in the type III secretion system in this experiment. *FliC* encodes a flagellar filament-type structure and can export proteins more efficiently after the hook structure.

The c-di-GMP does not regulate *fliC*, according to earlier research. However, when the *dgc* gene was suppressed, the production of *fliC* was lost, and the FliC protein could not be transmitted out of the cell to construct a complete flagellar structure, resulting in the inability to send low molecular weight bacteriocin out of the cell.

Furthermore, prior research has revealed that *fliC* and *flhA* mutants are unable to manufacture bacteriocins, but the two genes do not control each other, and *dgc* mutants have no influence on *flhA* expression [25], although the exact mechanism is unknown at this time. To regulate the *fliC* gene, what receptor does c-di-GMP bind to? When *clp* is altered in *Xcc*, the transcriptional expression of *fliC* is decreased, and the *clp* mutant produces no flagella and cannot move under electron microscopy. As a result, c-di-GMP and CLP will alter *fliC* transcription in *Xcc* [26], and more research into how c-di-GMP regulates *fliC* transcription in *Pcc* is required.

FliG is a switch protein for flagellar movement on the T3bSS inner membrane group’s flagellar organelle, which dictates the composition and rotation direction of the flagella and mostly controls flagella torsion [27]. The bacteriocins secretion was found to have no effect on the fliG mutant generated in the previous research. The c-di-GMP is the signaling molecule that predominantly impacts flagellar motility in many studies, and variations in c-di-GMP concentration cause bacteria to go from a planktonic state to a surface-attached state; therefore, this protein has little to do with the transport of bacteriocins [19].

In 2021, the *Pilz* gene was identified in the NCBI database’s *Pcc* gene. *Pilz* is a polysaccharide synthesis, cell motility, and cell differentiation c-di-GMP receptor protein. The YcgR protein from *Escherichia coli* is the best example of the PilZ proteins in the current investigation. The YcgR-c-di-GMP complex interacts with the flagellar motor protein FliG, which regulates flagellar rotation and allows swimming speed, in addition to the PilZ domain [28,29,30].

We know from the findings of this experiment that the *dgc* mutant prevents *fliG* transcription, indicating that c-di-GMP is still vital in controlling flagella movement in the *Pcc* strain. When the speed of bacterial migration increases, the ability to infect increases as well.

Bacteria’s ability to notice changes in the environment is crucial to their high adaptability to the environment. *Pcc* penetrates plants via lenticels or wounds and remains in the intercellular space until environmental circumstances are favorable for bacterial growth. The growth of bacteria, as well as the performance of virulence, is influenced by pH, carbon source, and nitrogen source. *Pcc* must absorb nutrients through protein channels on the membrane in order to survive in nature.

GlnH is an ATP-binding cassette-binding active transporter protein that is necessary for L-glutamine transport. It may be deduced from the action of various exogenous quantities of L-glutamine on 89-H-rif-12 that when L-glutamine surpasses a specific concentration, *crp* interacts with it. The downstream gene *dgc*, the third secretion system-related protein genes *flhA*, *fliC*, *fliG*, and the bacteriocin poisoning protein genes *caroS2k*, *caroS3k*, and *caroS4k* are all significantly expressed, and L-glutamine has been shown to alter the expression of virulence in other bacteria as well.

Other studies have found that because ammonia makes up the majority of nitrogen in soil and water and L-glutamine is scarce, Listeria cells with low ammonia and high L-glutamine will activate the transcription of virulence genes [31]. However, mammalian cells with extremely little ammonia and very high L-glutamine will promote virulence gene transcription [32,33].

When the concentration of L-glutamine in *Pcc* cells exceeds a particular level, it is thought that some proteins can detect favorable environmental conditions for development and send out signals to enhance CRP expression, altering motility and toxicity. Our findings support previous evidence indicating that L-glutamine, an abundant nitrogen source, significantly increases the expression of all main virulence genes in pathogens [34,35].

## 4. Materials and Methods

*Bacterial strains, plasmids, media, and growth conditions.* The bacterial strains, plasmids, and primers utilized in this study are listed in Table 2 and Table 3. *Pcc* strains were cultured on a modified Luria-Bertani (LB) medium with 5 g sodium chloride per liter at 28 °C (half of the recommended quantity of NaCl). Strains of *E. coli* (cloning host) were grown in LB broth at 37 °C with rotary agitation at 125 rpm. The IFO-802 medium was supplemented with 1% polypeptin, 0.2% yeast extract, 0.1% MgSO4 (pH 7.0), and 1.5% agar. Antibiotic concentrations employed in the bacterial selection, 50 g/mL ampicillin, 50 g/mL kanamycin, and 50 g/mL rifampicin, were used to treat *E. coli* and *Pcc* strains. All bacterial growth densities were measured at 595 nm using a spectrophotometer (OD_595_).

*Bacterial matting.* The membrane-filter method was used for bacterial mating, as described by Gantotti et al. [37]. *Pcc* (receiver) H-rif-8-6 and *E. coli* cultures overnight *E. coli* (donor) 1830 were equally spread on 0.22 µm pore size membrane filters (Millipore, Inc. Bedford, MA, USA) and incubated overnight at 28 °C on LB agar media. After conjugation, progeny suspensions were properly diluted and cultured on modified Drigalski agar plates (with rifampicin and kanamycin, 100 g ml^−1^) at 28 °C for 24 to 48 h. For the bacteriocin production test, colonies were isolated.

*Bacteriocin assays.* The antibacterial activity of the bacteriocin-producing strain was assessed using the soft agar overlay method [39]. Hard IFO-802 (1.4% agar) and soft IFO-802 (0.65% agar) medium were used to grow the isolates. For colonies to develop, the cells were first cultured for 12 h. After that, the colonies were subjected to ultraviolet light before being incubated for another 12 h. The cells were then treated with chloroform before being covered with soft agar that contained the indicator cells. Bacteriocin production is indicated by an inhibitory zone of indicator-cell (*SP33*) development surrounding the colony.

*Preparation of genomic DNA, plasmid DNA and RNA.* Standard protocols for restriction endonuclease digestions, agarose gel electrophoresis, purification of DNA from agarose gels, DNA ligation, and other cloning-related techniques were followed [40,41]. *E. coli* DH5α cells that are exponentially growing (OD_595_ of about 1.0) were extracted for RNA extraction. RNA was extracted using Trizol reagent (Invitrogen, Waltham, MA, USA) and resuspended in DEPC-treated water according to the manufacturer’s instructions. The purity and concentration of total RNA were measured using a NanoVue Plus^TM^ spectrophotometer (Biochrom, Holliston, MA, USA), followed by electrophoresis on a 1.5% formaldehyde-morpholinepropanesulfonic-agarose gel for evaluation.

*TAIL-PCR and restriction DNA library screening.* Detailed protocols have been utilized for the PCR, which used Promega’s Go-Taq DNA polymerase [42]. Thermal asymmetric interlaced PCR (TAIL-PCR) was utilized, as previously reported [41,43].

For TAIL-PCR, specific primers were developed at both ends of Tn5, namely PF1, PF2, PF3, PR1, PR2, and PR3. Two PCR products with unknown DNA sequences at both ends of Tn5 were produced using the above primers for amplification. However, after sequencing the two segments, it was discovered that they were the same PCR products. As a result, two sets of two PCR products were created utilizing this known 500-bp DNA sequence.

The TAIL-PCR products were sequenced using an ABI PRISM Dye Terminator Cycle Sequencing Ready Reaction kit (Applied Biosystems, Foster City, CA, USA). A GeneAmp System 9600 thermocycler was used to sequence the cycles (Applied Biosystems). According to the manufacturer’s procedure, the sequencing was conducted using an ABI 373S automated DNA sequencer 373S (Applied Biosystems).

*Subcloning of dgc gene from 89-H-rif-12.* Using the oligonucleotide primers *Pcc DGC* F upstream and *Pcc DGC* R downstream, the DNA fragment of *dgc* was amplified by PCR from 89-H-rif-12. TA cloning (Promega Inc., Madison, WI, USA) was used for subcloning the PCR product into the pGEM-T Easy vector, resulting in the plasmid pG-*dgc*. As previously described [35,38,39], SLIM excision of the tag element within the ribosome binding site and start codon of *dgc* in pG-*dgc* gave the construct pG-*dgc*-Kan. *DGC* Xho1 Fs, *DGC* Xho1 Ft, *DGC* Xho1 Rs, *DGC* Xho1 Rt, *DGC* Hpa1 Fs, *DGC* Hpa1 Ft, *DGC* Hpa1 Rs, and *DGC* Hpa1 Rt were used as primers. pG-*dgc*-Kan was then injected into *E. coli.* BL21 (DE3) cells were used.

Electroporation (1.25 kV/cm, 200, 25F) was used to insert plasmids into *Pcc* strains [44,45]. The competent E coli cells for heat-shock transformation were obtained using Hanahan’s method [33,37].

## 5. Conclusions

Under environmental stimulation, *Pccs* manufacture bacteriocin and defense protein molecules via the SOS mechanism. The c-di-GMP, which is formed from GTP by diguanylate cyclases, is a widely conserved bacterial signaling molecule that influences motility, biofilm formation, and virulence. This is the first study to show a link between *dgc* and the genes involved in *Pcc* LMWB extracellular export via the flagellar-type secretion system. The probable mechanism of *dgc* is hypothesized to synthesize the secondary signaling molecule c-di-GMP, which can bind to receptor proteins and positively influence bacteriocin transcription as well as toxin production, motility, and transport.

## Figures and Tables

**Figure 1 ijms-23-05649-f001:**
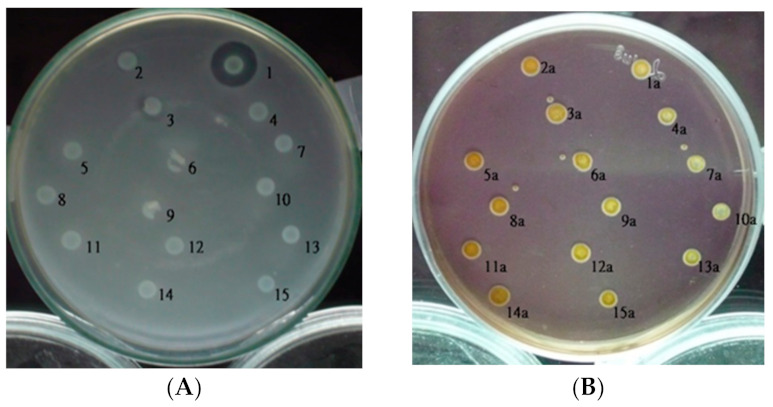
Bacteriocin of *Pectobacterium carotovorum* subsp. *carotovorum*. (**A**) The bacteriocin test was performed by soft agar overlay. (1) Wild-type rif-TO6 *Pcc* strain, (2–15) transposon Tn5 insertional mutants: 2—TT6-1, 3—TT6-3, 4—TT6-2, 5—TT6-4, 6—TT6-5, 7—TT6-6, 8—TT6-9, 9—TT6-12, 10—TT6-13, 11—TT6-14, 12—TT6-19, 13—TT6-21, 14—TT6-22, and 15—TT6-25. (**B**) Screening of the wild-type rif-TO6 and transposon Tn5 insertional mutants with Drigalski medium. If the strain is *Pcc*, the medium will be golden yellow, but if it is an *Escherichia coli* colony, the medium will be blue. (1) Wild Type rif-TO6 *Pcc* strain, (2–15) transposon Tn5 insertional mutants 2—TT6-1, 3—TT6-3, 4—TT6-2, 5—TT6-4, 6—TT6-5, 7—TT6-6, 8—TT6-9, 9—TT6-12, 10—TT6-13, 11—TT6-14, 12—TT6-19, 13—TT6-21, 14—TT6-22, and 15—TT6-25.

**Figure 2 ijms-23-05649-f002:**
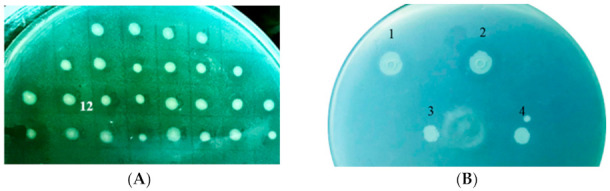
Bacteriocin assay of the gene knockout strains. (**A**) Bacteriocin activity of the gene selection strain (89-H-rif-12). (**B**) Comparison of bacteriocin activity of wild types and dgc defective mutants: 1, *Pcc* 3F3; 2, 89-H-rif-89-12; 3, 89-H-rif-12/Δdgc-Tet; 4, 89-H-rif-12/Δdgc-Kan.

**Figure 3 ijms-23-05649-f003:**
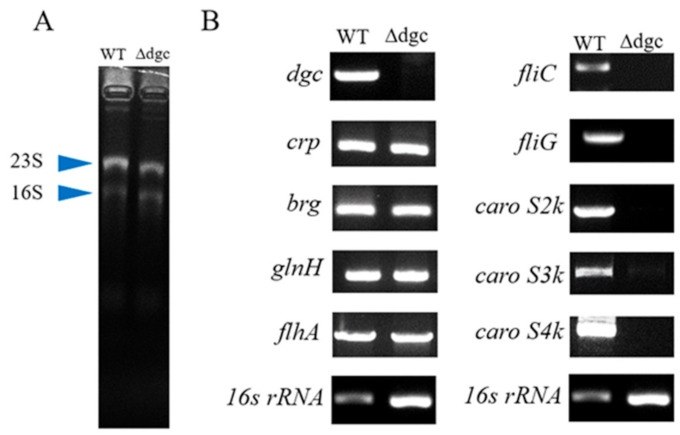
Transcriptional analysis of bacteriocin genes and regulatory genes by two-step endpoint RT-PCR. (**A**) 16s RNA expression of the wild-type 89-H-rif-12 and the dgc defective mutant. (**B**) Shown are gel electropherogram of RT-PCR products exhibited by the wild-type and dgc defective mutants.

**Figure 4 ijms-23-05649-f004:**
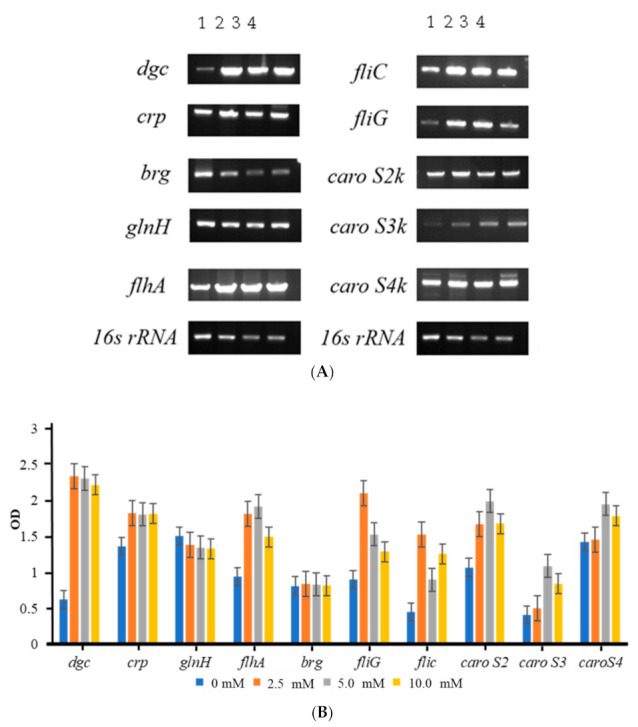
Transcriptional analysis of bacteriocin genes and regulatory genes by two-step endpoint RT-PCR induced with different concentrations of L-glutamine. (**A**) Gel electropherogram of RT-PCR products exhibited by the wild-type and the dgc knockout strain: 1, 0 mM; 2, 2.5 mM; 3, 5.0 mM; 4, 10 mM. (**B**) Mean relative gene expression affected by the L-glutamine expression.

**Figure 5 ijms-23-05649-f005:**
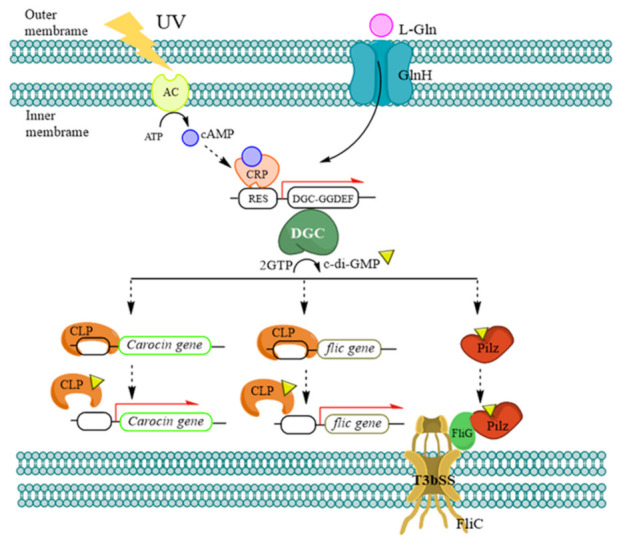
DGC and bacteriocin regulation, synthesis, and secretion: a proposed molecular pathway.

**Table 1 ijms-23-05649-t001:** List of genes identified by transposon mutagenesis.

Strains	Products
TT6-1	ATPase associated with various cellular activities, AAA_5, MoxR-like ATPase
TT6-2	bacteriophage tail protein [*Escherichia coli E22*]
TT6-3	peptidase S8 and S53 subtilisin kexin sedolisin; killer protein of pyocin S3
TT6-4	bacteriophage tail protein [*Escherichia coli E22*]
TT6-5	putative periplasmic ligand-binding sensor protein
TT6-6	diguanylate cyclase AdrA [*Shigella boydii Sb227*]
TT6-9	bacteriophage tail protein [*Escherichia coli E22*]
TT6-10	putative phage tail fiber protein [*Escherichia coli E24377A*]
TT6-13	killer protein of pyocin S3
TT6-14	putative bacteriophage protein GP48
TT6-19	putative phage tail fiber protein [*Escherichia coli E24377A*]
TT6-21	killer protein of pyocin S3
TT6-22	bacteriophage tail protein [*Escherichia coli E22*]

**Table 2 ijms-23-05649-t002:** Bacteria and plasmids used in this study.

Strain or Plasmid	Relevant Characteristics	Source of Reference
** *Escherichia coli* **		
DH5α	SupE44 ΔlacU169 (Φ80 lacZ ΔM15) hsdR17 recA1 endA1 gyrA96 thi-l relA1	Hanahan; Reusch et al. [33,36]
1803	pro^−^met^−^Km^r^Nm^r^, containing transposon Tn5 on the suicidal plasmid pBJ4JI	Gantotti et al. [37]
***Pectobacterium carotovorum* subsp. *carotovorum***		
TO6	*Pcc*, Amp^r^, wild type, putative biocontrol agent	Laboratory stock
Rif-TO6	*Pcc*, Amp^r^, Rif^r^, wild type	Laboratory stock
89-H-4	*Pcc*, Amp^r^, wild-type, putative biocontrol agent	Laboratory stock
89-H-rif-12	*Pcc*, Amp^r^, Rif^r^, wild type	Laboratory stock
3F3	*Pcc*, Amp^r^, wild type	Laboratory stock
SP33	*Pcc*, wild-type indicator strain	Laboratory stock
H-rif-89-12/∆*dgc-Kan*	*Pcc*, Amp^r^, Rif^r^, Kan^r^, wild type	This study
H-rif-89-12/∆*dgc-Tet*	*Pcc*, Amp^r^, Rif^r^, Tet^r^, wild type	This study
**Plasmids**		
pJB4JI	suicide vector for Tn5, pJB4JI, contains pPH1JI, bacteriophage Mu, and Tn5	P.R. Hirsch, J.E. Beringer, [38]
pGEM T-Easy	Amp^r^, lacZ cloning vector	Promega
pBR322	Ampr, Tetr	Bolivar et al. [21]
pG-*dgc*	pGEM T-Easy, Amp^r^, *dgc*	This study
pG-*dgc*-Xho1/Hpa1	pGEM T-Easy, Amp^r^, *dgc*	This study
pG-Tet	pGEM T-Easy, Amp^r^, Tetr	Laboratory stock
pG-Δ*dgc*-T	pGEM T-Easy, Amp^r^, *dgc*, Tet^r^	This study
pG-Δ*dgc*-K	pGEM T-Easy, Amp^r^, *dgc*, Kan^r^	This study

**Table 3 ijms-23-05649-t003:** Primers used in this study.

Primers	Sequence (5′-3′)
P3	CTCGACGTTGTCACTGAAGCGGGAAG
P4	AAAGCACGAGGA GCGGTCAGCCCAT
PF1	AGAGA ACACAGATTTAGCCCAGTCGG
PF2	CCGCACGATGAAGAGCAGAAGTTAT
PF3	GATCCTGGAAAACGGGAAAGGTTC
PR1	GCCGAAGAGAACACAGATTTAGCCCA
PR2	CCGCACGATGAAGAGCAGAAGTT
*Pcc*_*DGC*_F_upstream	CTCACTGTTGCTGACATGC
*Pcc*_*DGC*_R_downstream	ATTCAGGCAACTTCGGTTC
*DGC*_Xho1_Fs	CGACAATCCGTGGAATATAG
*DGC*_Xho1_Ft	CTCGAGCGACAATCCGTG
*DGC*_Xho1_Rs	GTACGATACTGTGCGCTC
*DGC*_Xho1_Rt	CTCGAGGTACGATACTGTG
*DGC*_Hpa1_Fs	GATAAGCTCTCCCGAATACG
*DGC*_Hpa1_Ft	GTTAACGATAAGCTCTCCCG
*DGC*_Hpa1_Rs	GCTGAAGTTTCTGAACCAG
*DGC*_Hpa1_Rt	GTTAACGCTGAAGTTTCTGAAC
F-16s- sense-ECC	CTGGACAAAGACTGACGCTC
R-16s- sense-ECC	TCGCTGGCAACAAAGGATAAG
caroS2K_RT_F	GAGATACAATGACCGTGGATGG
caroS2K_RT_R	GCAACTGGTGTTACCGTAACTG
caroS3K_RT_F	ATGATTAAGTACCGTTTATATGCTC
caroS3K_RT_R	TCATTGCGACTCCCTCATAT
CaroS4KI_RT_R	GGATCCATGATTAATTTTAAGG
CaroS4KI_RT_R	GAGCTCTTAGAGACCGTAT
CRP_RT_F	CTCTCGAATGGTTCCTTTCC
CRP_RT_R	GAGATCAGGTTCTGGTCTTC
FlhA_RT_F	TCACTCAAGCTTGCATCTAC
FlhA_RT_R	AAGCTTTCACTTCTGAGCTTCC
GlnH_ RT_F	ACAGACCGGTGAATTACGCATCGG
GlnH_ RT_R	GCCGCTACGCCTTCATCCATATTC
FliG sense	ATGACCCTGACAGGAACAG
FliG antisense	TTAGACATAAGCATCCTCGC
DY-F1	GGTAGGATCCGTTGTTAGGTGCATAGGTTGG
DY-R1	TTCAAGCTTGTGGTGAATTGACAATACGC

## Data Availability

Not applicable.

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
