# Peer review of "Diguanylate Cyclase (DGC) Implicated in the Synthesis of Multiple Bacteriocins via the Flagellar-Type III Secretion System Produced by Pectobacterium carotovorum subsp. carotovorum"

_ijms, 2022, doi:10.3390/ijms23105649_

Round 1
Reviewer 1 Report
Dear Authors, your manuscript about Pectobacterium carotovorum subsp. carotovorum and the ability to synthetize of bacteriocins via flagellar type three secretion system is interesting, in particular for plant pathologists.
You might implement the introduction with some informations about this point: In many bacterial species, etc.: could you introdure what bacterial species? and about the pathogenicity of the bacterium
Moreover, in Figure 5 (lines 210-211) why you indicate UV? please explain
line 266: the sentence related to Listeria is not cleare in this point: is it an example?
Please put in italic all the names of bacteria (lines 70, 76, 81, etc)
Please put in italic all the genes (lines 93, 149, 152, 180, 181, 188, 221, etc )
line 131 put "in vivo" in italic
line 140 put "in vitro" in italic
Reviewer 2 Report
The manuscript titled “Diguanylate cyclase (DGC) implicated in the synthesis of multiple bacteriocins via the flagellar type three secretion system produced by Pectobacterium carotovorum subsp. carotovorum” is devoted to Pcc's virulence factor – DGC that cyclizes c-di-GMP, a common secondary signal in physiological processes and toxin control systems of many bacteria. In this study the dgc gene in Pcc was blocked using the method of homologous recombination. In the in vivo setting, the results demonstrated that the dgc knock-out strain does not release low molecular weight bacteriocins. The bacteriocin gene (carocin S2, carocin S3, carocin S4) and the flagellar type three secretion system genes were also unable to be transcribed by the dgc knock-out strain in the transcription experiment. We also observed that the amount of bacteriocin expressed changed when the amount of L-glutamine in the environment exceeded a particular level. These data suggested that L-glutamine influenced physiological processes in Pcc strains in some way. Rrelationship between dgc and the genes involved in Pcc LMWB external export via the flagellar type secretion system was proposed based on these findings. DGC's cyclic di-GMP might bind to receptor proteins and positively regulate bacteriocin transcription as well as the synthesis, mobility, and transport of toxins.
The manuscript reports important information and describes carefully made research, but it still need additional work to improve English and logic of text, especially in introduction and discussion parts. The sence of many sentences is not clear, so far, some misunderstanding is possible.
Some particular notes:
Line 32 When bacteria are activated by their surroundings, they create a number of transmission pathways in their cells that cause physiological changes involving many proteins and chemicals.
- Please, be more precise and specific.
Line 44 There are several types of c-di-GMP receptors, including the PilZ protein domain, which affects both animals and plants.
- DGC do not “affects both animals and plants”. DGC plays important role in physiology and biochemistry of both animals and plants.
Line 47 Recent findings suggest that c-di-GMP affects type VI secretion systems (T6SSs)[14] as well as T3SSs [15].
-Citation is not correct: the reference 14 is describing both switching of Type III And Type VI Secretion Via C-Di-GMP Signalling.
14) Moscoso, Joana A., Helga Mikkelsen, Stephan Heeb, Paul Williams, and Alain Filloux. 2011. "The Pseudomonas Aeruginosa Sensor Rets Switches Type III And Type VI Secretion Via C-Di-GMP Signalling". Environmental Microbiology 13 (12): 3128-408 3138. doi:10.1111/j.1462-2920.2011.02595.x.
Line 51 Pectobacterium carotovorum subsp. carotovorum (Pcc) secretes numerous pathogenicity-related characteristics that cause soft-rot disease in a variety of plant species [16].
-Please, change “secretes numerous pathogenicity-related characteristics” for “secretes numerous pathogenicity-related proteins”
Line 104 Table 1 TT6-6 diguanylate cyclase AdrA [Shigella boydii Sb227] contradicts to the Line 108: “Among the different insertion mutants, the TT6-6 DNA sequence revealed that this carries a diguanylate cyclase with a 39% homogeinety in Serratia, according to results from the BLAST programs of the National Center for Biotechnology Information website 110 (National Library of Medicine, USA) Table 1.
Please, chose one most similar protein for comparison (Shigella boydii Sb227 or diguanylate cyclase in Serratia).
Line 113 We used rifampicin-resistant Pcc 89-H-12 resistant bacteria as the gene knock-out strain to better understand the link between DGC and bacteriocin production
- We used rifampicin-resistant Pcc 89-H-12 bacteria…
Line 121 Because the 89-H-12 strain can manufacture carocin S2, S3 and S4, the low-molecular-weight bacteriocins (Figure 2A), using it as a gene selection strain will help the researcher better understand Pcc's virulence mechanism.
- It is better to use another word instead of manufacture (which mean more production of goods)
Line 256 Pcc must absorb nutrients through protein channels on the membrane in order to survive in nature. As a result, L-glutamine must rely on L-glutamine for transportation. GlnH is an active transporter protein that binds to ATP-binding cassettes.
-Please, re-write the sentences for clear sence.
Line 271. The mechanism by which L-glutamine transfers virulence genes in these pathogens, as well as the universality of this mechanism across pathogens, remains unknown.
- L-glutamine does not transfer virulence genes, it can modify the gene expression.
- Role and possible mechanism of L-glutamine modulation of virulence genes expression is described in a number of reports, including
Portman JL, Dubensky SB, Peterson BN, Whiteley AT, Portnoy DA. Activation of the Listeria monocytogenes virulence program by a reducing environment. MBio. 2017 Oct 17;8(5):e01595-17.
Haber A, Friedman S, Lobel L, Burg-Golani T, Sigal N, Rose J, Livnat-Levanon N, Lewinson O, Herskovits AA. L-glutamine induces expression of Listeria monocytogenes virulence genes. PLoS pathogens. 2017 Jan 23;13(1):e1006161.
Minato Y, Fassio SR, Wolfe AJ, Häse CC. Central metabolism controls transcription of a virulence gene regulator in Vibrio cholerae. Microbiology (Reading). 2013 Apr;159(Pt 4):792-802. doi: 10.1099/mic.0.064865-0. Epub 2013 Feb 21.
This part of discussion must be considerably changed and improved.
